# Development of SNP Set for the Marker-Assisted Selection of Guar (*Cyamopsis tetragonoloba* (L.) Taub.) Based on a Custom Reference Genome Assembly

**DOI:** 10.3390/plants10102063

**Published:** 2021-09-30

**Authors:** Elizaveta Grigoreva, Yury Barbitoff, Anton Changalidi, Dmitry Karzhaev, Vladimir Volkov, Veronika Shadrina, Elizaveta Safronycheva, Cécile Ben, Laurent Gentzbittel, Elena Potokina

**Affiliations:** 1Institute of Forest and Natural Resources Management, Saint Petersburg State Forest Technical University, St. Petersburg 194021, Russia; L.Grigoreva@gmail.com (E.G.); karzhaevd@gmail.com (D.K.); vol-j@mail.ru (V.V.); ver3301@yandex.ru (V.S.); esafronycheva@mail.ru (E.S.); 2Sirius University of Science and Technology, Sochi 354340, Russia; 3Department of Cytology and Histology, Saint Petersburg State University, St. Petersburg 199034, Russia; barbitoff@bioinf.me; 4Department of Genetics and Biotechnology, Saint Petersburg State University, St. Petersburg 199034, Russia; 5Bioinformatics Institute, St. Petersburg 197342, Russia; anton.chana@gmail.com; 6Faculty of Software Engineering and Computer Systems, ITMO University, St. Petersburg 197101, Russia; 7Skolkovo Institute of Science and Technology, Moscow 121205, Russia; C.Ben@skoltech.ru (C.B.); L.Gentzbittel@skoltech.ru (L.G.)

**Keywords:** guar, RADseq, SNPs, population structure, GWAS, *myo*-inositol phosphate metabolism

## Abstract

Guar gum, a polysaccharide derived from guar seeds, is widely used in a variety of industrial applications, including oil and gas production. Although guar is mostly propagated in India, interest in guar as a new industrial legume crop is increasing worldwide, demanding the development of effective tools for marker-assisted selection. In this paper, we report a wide-ranging set of 4907 common SNPs and 327 InDels generated from RADseq genotyping data of 166 guar plants of different geographical origin. A custom guar reference genome was assembled and used for variant calling. A consensus set of variants was built using three bioinformatic pipelines for short variant discovery. The developed molecular markers were used for genome-wide association study, resulting in the discovery of six markers linked to the variation of an important agronomic trait—percentage of pods matured to the harvest date under long light day conditions. One of the associated variants was found inside the putative transcript sequence homologous to an ABC transporter in *Arabidopsis*, which has been shown to play an important role in D-*myo*-inositol phosphates metabolism. Earlier, we suggested that genes involved in *myo*-inositol phosphate metabolism have significant impact on the early flowering of guar plants. Hence, we believe that the developed SNP set allows for the identification of confident molecular markers of important agrobiological traits.

## 1. Introduction

Guar or cluster bean (*Cyamopsis tetragonoloba* (L.) Taub.) is an annual legume crop that has been used by the local populations of India and Pakistan for food and as feed for cattle, providing a good source of nutrition both for humans and animals. Guar was further distributed around the world in the middle of the last century due to the discovery of a valuable polysaccharide—galactomannan (guar gum)—in its seed endosperm. Due to its ability to form gels with cold water, guar gum has found wide application in the food, textile, and cosmetic industries. However, the main demand for guar gum comes from the oil and gas companies, which are using guar-based fluids to improve the efficiency of hydraulic fracturing. India is responsible for ~80% of total guar production in the world (e.g., 2.7 million tons of guar gum was generated in India during the 2013–2014 agricultural year) [1]. Guar is currently being introduced as a new crop in other countries, including the USA, Italy, Morocco, Germany, Spain [2], and, more recently, Russia [3].

The introduction of guar as a new legume crop to the Western agriculture requires significant breeding efforts aimed at the development of new high-yield varieties well adapted to new environments. Marker-assisted selection (MAS) based on the latest sequencing technologies can be effectively used to obtain disease/pest resistant guar cultivars with lower sensitivity to photoperiod and higher yield. However, genomic resources are limited in guar, and ‘omics’ technologies are rarely applied for this species [3].

The limitation is now gradually being overcome due to the recent transcriptomic studies of guar [4,5]. An RNA-Seq study of root tissues of two Indian guar cultivars (‘RGC-1066′ and ‘M-83′) identified a set of 25,040 high-confidence molecular markers, including 18,792 simple sequence repeats (SSRs), 5,999 single-nucleotide polymorphisms (SNPs), and 249 insertion/deletion variants (InDels) [6]. Another RNA-Seq-based transcriptome analysis of leaf, shoot, and flower tissues of the ‘RGC-936′ variety identified a different set of 8,687 potential SSRs, which were further embedded into the ‘ClustergeneDB’ database that allows user to retrieve information about cluster bean unigenes [5]. Yet another large set of 15,399 microsatellite marker-pairs was designed based on the results of next generation DNA sequencing of the ‘GG-4′ guar variety [7]. Finally, transcriptomes of three guar cultivars (‘M-18′, ‘RGC-1066′, and ‘RGC-936′) were used for short genetic variant discovery [8] using the unpublished draft genome assembly of the ‘RGC-936′ cultivar [9]. However, none of the aforementioned studies provided a detailed analysis of the huge intraspecific diversity of *C. tetragonoloba.* Such an analysis is important for the breeding and introduction of guar to various countries, as valuable rare alleles can be found in such a study. Genotyping of SNPs and InDels in a large set of guar samples of different origin might open the way for genome-wide association studies.

In the present study we aimed to construct a comprehensive set of SNPs and InDels by genotyping of 166 diverse guar plants introduced to Russia over the past few years from different sources. Our set of samples mainly comprises landraces brought in by breeders and industrial companies from India and Pakistan, as well as guar cultivars raised in the USA. Since a high-quality reference genome assembly of guar [9] is still not publicly available, we constructed a custom assembly using both short- and long-read sequencing technologies. As a genotyping method, we employed genotype-by-sequencing restriction site-associated DNA sequencing (GBS RAD-Seq). The high capacity and cost benefits of RADseq for SNP discovery are currently well acknowledged [10]. RAD-seq outperforms other molecular marker techniques (e.g., microsatellites) in applications that require individual-level genotype information [11] and is capable of identifying thousands of molecular markers that are polymorphic in large populations. Moreover, RAD-Seq is commonly applied even in the absence of a reference genome sequences. Various crops have been optimized by GBS in an efficient, high-throughput, and cost-effective manner [12]. Given these benefits of GBS RAD-Seq, it seems a promising approach to investigate the intra-species diversity of guar, complementing previous Illumina and Oxford Nanopore based attempts of SSR and SNPs discovery for this crop [6,13].

## 2. Results

### 2.1. Construction of a Wide-Ranging SNP Dataset for Guar

To construct a comprehensive set of SNPs, we performed the genotype-by-sequencing restriction site-associated DNA sequencing (GBS RAD-Seq) of guar plants belonging to 45 accessions of different geographic origin (Appendix A Appendix A). For the SNP calling procedure, two RAD-Seq libraries, each containing 96 samples with unique barcodes, were sequenced using the Illumina HiSeq2500 platform (150 bp single-end reads were generated). Each of the 96 barcodes in a run was sequenced on two lanes. In total, ~ 500 Mbp of high-quality Illumina reads were generated for all samples. The median count of reads per sample after demultiplexing varied from ~280,272 up to ~871,984, depending on lane/run. The median alignment rate varied from 86% to 99% depending on the sample and read alignment software.

Due to the incompleteness of the reference genome, as well as due to the high genotyping error rates in RAD-Seq [14], we decided to apply several software tools for variant discovery. We applied a general purpose variant discovery pipeline based on the Genome Analysis Toolkit’s Haplotype Caller (GATK-HC) [15], a solution that was shown to provide higher sensitivity for reduced representation sequencing at the cost of a higher error rate [16,17], and two dedicated GBS solutions: NGSEP [18] and TASSEL5 [19] (see Materials and Methods for details on each pipeline). We observed a remarkably low concordance of variant calls produced by different software tools (Appendix A, see Methods); hence, we decided to combine the results and select variants that have been discovered by at least two of the three tools (see Methods for details). The aggregation of the variant calls followed by subsequent quality control and MAF filtering resulted in a set of 5234 high-quality common variant sites (Figure 1).

### 2.2. Genotype Data Provide Insights into the Genetic Structure of the Guar Populations

We next used our constructed genotype data to obtain insights into the genetic structure of the guar population. To do so, we first used principal component analysis (PCA) to assess the general stratification of the population. Indeed, we observed that samples formed several distinct clusters in the space of the first two principal components (Figure 2a); at the same time, these clusters did not coincide with the known information about the accessions, including their country of origin (Figure 2a). These observations suggest that present-day varieties of guar did not yet accumulate a sufficient number of genetic changes to appear as separate lineages, possibly due to a recently started artificial selection. To validate this assumption, we also employed the genetic clustering algorithm, ADMIXTURE [20], that can be used to find a set K ancestral population and their relative contribution to the present-day samples. The results of such genetic clustering confirmed PCA-based observations and showed that plants coming from the same place of origin do not belong to the same cluster; at the same time, despite the clumping of some samples on a PCA plot, no clearly distinct evolutionary lineages are present in our reference set of samples.

Selfing may also contribute to the observed patterns of genetic variation and to the lack of clear separation between known varieties and geographical populations. High levels of selfing in a population increases the homozygosity of the individuals, which can be assessed directly using genotype frequencies. Indeed, we observed that the relationship between the non-reference allele frequency and the number of heterozygous genotypes in our dataset was significantly altered in our data compared to the Hardy–Weinberg equilibrium (Appendix A). For all allele frequencies, much fewer heterozygous calls were obtained compared to the HWE expectation. In concordance with these findings, estimation of the inbreeding coefficient (F) for each site in our common variant dataset showed high excess of homozygous genotypes, corresponding to high positive value of F (Figure 2c). While increased homozygosity of samples might also reflect the nature of the sample (a mixture of samples of different origin rather than a random sample from a single population), the genotype distribution confirms known aspects of guar biology and further confirms the general accuracy of genotyping.

In addition to its effects on genotype distribution, selfing also results in increased linkage disequilibrium between variants. Expectedly, we found that the estimated pairwise r^2^ between variants in our dataset decayed slowly with increasing the physical distance between markers, with a median r^2^ = 0.17 even for pairs of variants located at a physical distance of 1 million base pairs (1 Mbp) (Figure 2d, Appendix A). Taken together, these results show that selfing contributes to the patterns of genetic variation in guar.

### 2.3. Genome-Wide Association Analysis for Productivity-Related Traits

We next went on to identify the loci associated with productivity-related traits in guar using the constructed genotype dataset. Several traits were selected for this analysis, including the plant height, the height of the first branch attachment, the total number of pods, the percentage of pods matured to the harvest date, and the total weight of seeds harvested from mature pods. As the raw trait distributions were significantly skewed, the outliers for all traits were removed, and the raw values were normalized prior to genome-wide association analysis (see Methods). A moderate correlation between normalized trait values was observed, with maturation-related traits such as the number of pods, percentage of mature pods, and seed weight from mature pods forming a densely correlated cluster (Appendix A).

We applied two types of association analysis models to detect genome-wide association across our panel of phenotypes (e.g., plant height, height of the first branch attachment, total number of pods, percentage of pods matured to the harvest date, and total weight of seeds harvested from mature pods). GLM-based association testing did not identify any significant genome-wide associations (Appendix A); however, significant genomic inflation was observed in the association statistics only for one of the traits (percentage of pods matured to the harvest date) despite population stratification being taken into account during association testing. This observation led us to hypothesize some additional relatedness-driven factors influenced the GLM results. To better control for random genetic effects arising from kinship, we applied an iterative fixed and random effect model FarmCPU [21] for the association analysis. This approach, in contrast to GLM, identified a strong genome-wide association signal for the percentage of mature pods per plant by harvest date (Figure 3a) but not the other traits.

We next selected loci showing the association with the percentage of mature pods at *p* < 1 × 10^−4^. The GWAS hits were curated to exclude repetitive sequences or variants with no adjacent transcript matches. As a result, six variants were found corresponding to six different contigs in the reference genome assembly (Table 1). Out of the six selected variants, only one directly fell inside the regions matching the reference transcriptome assembly. The corresponding reference transcript encodes a protein homologous to an ABC transporter G family member-39 protein of *Glycine max* and *G. soja* as indicated by a BLAST search. For all other variants, we selected the reference transcripts matching regions closest to each of the variants. The corresponding transcripts were then analyzed using BLAST to obtain their functional annotation using the homologous *G. max* genes. The set of transcripts located near the associated variants corresponded to the *MYB20* transcription factor, the chloroplast *SECA2* protein transporter, the glycine cleavage system protein, the F-box/LRR-repeat protein, and the putative glucuronosyltransferase *PGSIP7*/8 (Table 1). Unfortunately, a Gene Ontology term analysis using the SoyBase database did not identify any significantly overrepresented terms among these top hits.

### 2.4. PCR Validation of the Bioinformatics-Based SNP Prediction

The in silico analysis of the GBS RAD-Seq data described the above-identified 4907 SNPs and 327 InDels (Appendix A), 6 of which showed significant genome-wide association with the percentage of pods matured to harvest date (Table 1). As there are usually many pitfalls in this bioinformatic data analysis and SNP calling (especially for GBS data), independent confirmation of the reliability of the developed markers is always required.

To provide such an independent confirmation, we first performed the direct experimental validation of the identified markers, focusing primarily on the ones associated with the variation of early maturing of guar plants (Table 1). Firstly, we extracted the up- and downstream sequences flanking each of the six polymorphic loci from Table 1 in the custom reference genome assembly. Flanking genomic sequences allowed us to easily design locus-specific primers for experimental validation (Appendix A Appendix A).

Next, using the genotype information obtained during in silico analysis (Appendix A Appendix A), we focused on a subset of 12 guar genotypes harboring a maximal number of alternative alleles at the 6 loci studied (s19, s26, s33, s36, s51, s59, s73, s81, s109, s110, s168, s170). The subset of plants for experimental marker validation was designed to include plants with important rare alleles. For example, for the SNP located at the 3026232th position in the ‘Contig_188′ sequence, the ‘A’ allele in the homozygous state was predicted for 152 guar plants out of 166 studied, whereas the alternative ‘C’ allele was expected for only 9 plants. So, genotypes s36 and s110 carrying the rare ‘C’ allele of the SNP were included in the verification subset of genotypes (Appendix A).

Using the DNA samples of the 12 selected plants (the same samples were used to prepare the RADseq library), we performed PCR with the designed locus-specific primers. PCR products were directly used for Sanger sequencing. Sanger sequencing confirmed the physical presence of the target nucleotide substitutions in a subset of 12 plants (Appendix A), in a perfect concordance with the results of the bioinformatic data analysis (Appendix A). In addition to the target SNP and InDel, some additional SNPs genetically linked to the target ones were identified, forming candidate haplotypes.

Once we confirmed the physical presence of SNPs in the guar genome sequence, we proceeded to validate the effects of variants listed in Table 1 on the early maturity trait variation in the studied population. Interestingly, not all variants showed significant effects on the trait in the simple between-group comparison with the Wilcoxon rank sum test. However, the most significant difference in the percentage of pods matured to the harvest date was expectedly found between the allele classes of the InDel marker named Contig_59:14389378, which also showed the most significant *p*-value, determined by the Fixed and Random effect model (Table 1). For homozygotes A/A, the mean value of mature pods was 80 ± 1.1%, while homozygotes carrying a single nucleotide ‘G’ insertion at the locus (AG/AG) had only 65 ± 3.0% of pods matured to the harvest date. For heterozygotes (A-/AG), the mean value of the trait was 71 ± 5.9%.

Importantly, the only possibly functional SNP ‘Contig_182: 841660’ found in the coding sequence of the ABC transporter gene (Table 1) also showed a very significant difference in the average percentage of mature pods between genotypes with alternative alleles: T/T—79 ± 1.2%; C/C—66 ± 3.1%; and T/G—77 ± 3.5%. These two markers can be recommended for a marker-assisted guar breeding program.

## 3. Discussion

High-density SNP markers derived from RAD-seq data expand opportunities for the analysis of intraspecies genetic diversity and open the way for genome-wide association studies. Here, we report a comprehensive polymorphic loci dataset, combining 4,907 SNPs and 327 InDels, generated from RADseq genotyping data of 166 guar plants. Despite a large number of samples, we identify fewer high-quality variants compared to previous studies, including a recent analysis of just two guar varieties ‘RGC-1066’ and ‘M-83’ based on the RNA-Seq analysis [6]. This observation may be explained in two ways: first, our dataset is limited to variants that are common in the population and does not include extremely rare variant sites; second, bioinformatic pipelines heavily influence the variant calling results, offering a choice between the quantities of detected markers and their quality.

The robustness of the developed SNP set was confirmed by classic PCR-based assay followed by Sanger sequencing of the PCR product (see above). In addition to the successful experimental validation of the selected variants, we consider the moderate level of heterozygosity (mean value of 11%) observed in the estimated population of this obligatorily self-pollinated species as indirect evidence of the reliability of markers. In a typical RAD library, millions of reads are generated and then allocated to each multiplexed individual and to each genomic position in the reference genome [22]. The incompleteness of the reference genome often leads to improper stacking of raw reads belonging to paralogous loci or originating from repeat regions. This results in a false excessive heterozygous discovery. Although the maximal heterozygosity up to 50% was allowed by filtering procedure in our analysis, 90% of discovered SNP showed the heterozygosity percentage less than 25% with a median value of 8% (Appendix A).

The genotyping of the diverse guar samples stored in a large ex situ collection showed few to no correlations between the origin of accessions and their genetic variation profiles. While the possibility of errors during sample collection cannot be ruled out completely, plants from the same accessions (or even assigned to the same cultivar) were often observed in different genetic clusters. These results are consistent with previous reports about the discrepancies between phenotype-based and molecular-based clustering results in guar [23] and also with reports about genetic clustering of guar genotypes, irrespective of their geographical origins [24].

The possible reason for the lack of the obvious evolutionary lineages of guar is the relatively recent breeding history of the crop. It is frequently considered that guar has been domesticated in the northwestern region of the Indo-Pakistan subcontinent [25], and, as a center of domestication, this region harbored the highest genetic diversity of the species. Guar was introduced into the USA in 1903 [26], but efforts on the genetic improvement of guar in this region were limited until 1985, when the ‘Santa Cruz’ and ‘Lewis’ cultivars were released. Two other improved American guar cultivars (‘Matador’ and ‘Monument’) and several hundred highly diverse breeding lines were developed in 1998–2007 in the program of conventional plant improvement, starting with an evaluation of almost 200 plant introductions from the USDA-ARS collection at Griffin, Georgia [27]. Likewise, the guar breeding program in Russia started only in 2018 when the first cultivars (e.g., Vavilovskij 130), selected from new entries of the VIR collection, were officially registered. Such a small evolutionary timescale is not sufficient for the distinct lineages to emerge in different breeding regions.

The development of the set of SNPs provides an opportunity for the further genome-wide association study aimed at finding the genetic loci linked to variation of agrobiological traits. The prolonged vegetation period of plants upon their introduction to countries of more northern latitudes presents a major problem for guar breeding. This causes a delay in the harvest before the onset of autumn rains and negatively affects the yield [3]. Thus, the earliness of the pods’ maturation assessed as a percentage of pods matured to the harvest date is of the greatest importance among the yield-contributing traits. Fortunately, a strong genome-wide association signal (six variants with *p* < 0.0001, Table 1) was discovered for this key agrobiological trait.

Using the genome-guided guar transcriptome assembly, that was based on RNA-Seq data from 15 plants included in the present analysis [28], we were able to investigate whether the associated SNPs directly correspond to the coding sequence or regulatory regions. Among the six SNPs associated with the percentage of pods matured to the harvest date, just one (contig_182:841660, Table 1) was found inside the putative transcript sequence. The corresponding transcript was homologous to an ABC transporter G family member-39 protein of *Glycine soja* and to ATP-BINDING CASSETTE G40 (*AT1G15520*) in *Arabidopsis*. Earlier, we detected increased expression of the same transcript in early flowering guar plants grown under long day conditions compared with delayed flowering plants [28]. Importantly, the early-flowering guar genotypes were also proven to be early maturing, showing the highest percentage of matured seeds by harvesting (‘maturity index’) [3]. Therefore, a direct relationship can be assumed between the increased expression level of the *AT1G15520* orthologous gene of guar and early maturing of guar plants. Since the structure of the corresponding gene in guar has not yet been investigated, there is no option to predict any amino acid substitutions caused by the discovered SNP. Nevertheless, we can assume the presence of tight linkage disequilibrium between SNPs in the gene exon and other polymorphisms in its promoter region that affect gene expression.

An ABC transporter gene (*AT1G15520*) was recently shown to play a significant role in D-myo-inositol phosphates (IPs) metabolism in *Arabidopsis* [29]. The latter presumes the sequential reversible phosphorylation of the 6-carbon cyclic alcohol *myo*-inositol ring (Ins) resulting in the synthesis of a series of soluble Ins phosphates, as follows: Ins → Ins(3)P1→ Ins(3,4)P2 → Ins(3,4,6)P3 → Ins(1,3,4,6)P4 → Ins(1,3,4,5,6)P5 → Ins(1,2,3,4,5,6)P6 [30]. It is noteworthy, that perturbations in the IPs metabolic pathways, leading to the increased concentration of *myo*-inositol in its free dephosphorylated form, was previously suggested as a biomarker of early flowering (and presumably early maturing) of guar plants [28]. In maize, the ABC transporter-like gene (orthologous to *AT1G15520* in *Arabidopsis*) was co-expressed with the purple acid phosphatase (PAP)-like gene [29], whose ortholog in *Arabidopsis* (*AtPAP15*) encodes phytase that hydrolyzes *myo*-inositol hexakisphosphate (Ins(1,2,3,4,5,6)P6, also known as phytic acid) to yield free *myo*-inositol and free phosphate [31]. These data allow us to hypothesize that the aforementioned SNP in the guar genome affects pod maturation by affecting the IP metabolic pathways. However, the details of this mechanism deserve further investigation.

Two other associated SNPs (contig_59:14389378 and contig_188:3026232) were discovered near the guar coding sequences, showing homology to transcription factor *MYB20* and F-box/LRR-repeat protein 3, respectively (Table 1). Remarkably, a possible role of these proteins in metabolic pathways of IPs was also revealed in maize. Twenty *MYB* genes were sustainably differentially expressed in embryos of two maize inbred lines with significant differences in phytic acid content [29]. F-box protein was found to be a possible regulator of Phospholipase C (*PLC*) gene in the maize embryos, it was proposed that small molecules, e.g., auxin, can bind to F-box protein and mediate ubiquitination to regulate protein function at the posttranslational level [29]. Phospholipase C is involved in Ca^2+^ signal transduction [32] and IP metabolism by releasing IP3 from phosphatidylinositol [33]. We recently found that the phosphatidylinositol signaling pathway is initiated in early-flowering guar genotypes through the activation of the phospholipase C (*PLC*) gene, resulting in an exponential increase in the amount of *myo*-inositol in its free form [28]. Thus, function annotation of the contig_188 sequence may reveal genes playing key roles in earliness of guar varieties.

Overall, our results demonstrate the utility of the constructed set of SNPs and InDels for the analysis of genome-wide associations in guar and for the reconstruction of its population history. We believe that this dataset might be used for development of the genotyping methods and for further investigation of the guar genetics.

## 4. Materials and Methods

### 4.1. Sample and Phenotypes Collection

Guar plants investigated in the present study were derived from 45 accessions of the collection of Vavilov Institute of Plant Genetic Resources (VIR) (Appendix A Appendix A). Some of the accessions contained very diverse genotypes, while other accessions were represented by morphologically uniform plants. Hence, between 1 and 18 plants per accession were taken for analysis. The plants with the most contrasting phenotypes were selected individually during reproduction of the accessions in field experiments in the Krasnodar area of Russia (45°02′55″N). During the experiments each plant was assessed for several yield-contributing traits, e.g., plant height, number of pods, percentage of pods matured to the harvest date, weight of seeds from mature pods, and also for the height of attachment of the first branch (this trait is important for mechanical harvesting). Seeds were collected from each selected plant individually and later germinated under greenhouse conditions. The first true leaf of one seedling per plant was used for DNA extraction and subsequent RADseq genotyping.

### 4.2. RAD-Seq Library Preparation and Sequencing

Procedure of DNA isolation for RADseq genotyping as well as protocol for construction and sequencing of ddRAD libraries were described earlier [34,35]. Sequencing of ddRAD libraries was performed on Illumina HiSeq2500 with single end reads of 150 base pairs.

### 4.3. WGS Library Preparation and Custom Reference Genome Assembly

For the SNP calling procedure, the custom guar reference genome was employed. For its construction, DNA from leaves of guar variety ‘Vavilovskij 130′ was used [36]. Two different sequencing technologies were used for the reference genome assembly: short reads generated by Illumina NovaSeq6000 and long reads obtained by Oxford Nanopore Technologies (ONT). Short reads library was prepared following standard Illumina WGS protocol and sequenced on two lanes (2×150 mode) of the Illumina NovaSeq6000 sequencer. Long reads libraries for minION sequencing were constructed according to SQK-LSK 109 ONT protocol.

Short and long reads data were preprocessed: base calling with included step of removing low quality data was performed for minION by Guppy v. 0.1.11. Then, for both types of data (short and long reads) the most common contaminants and chloroplast DNA were removed using Bbmap v. 38.62, low quality reads were trimmed by Trimmomatic v. 0.38 for Illumina.

For the downstream assembly, different strategies were applied and compared: long-reads assembly, short reads assembly, and hybrid assembly combining short and long reads. For the short read assembly, Abyss v. 2.2.5, Soapdenovo2 v. 2.2.4, and SGA v. 0.10.13 tools were used. For the long reads assembly, Flye v. 2.8. and Canu v.1.9. were employed with additional polishing step using Racon v. 1.4.11. For the hybrid assembly we used MaSuRCA v.3.3.7., Abyss v. 2.2.1, and Spades (hybridspades) v.3.13.2. tools.

All of the assemblies were compared using basic assembly metrics via Quast v. 5.0.1. As a result, MaSuRCA-derived assembly was chosen as the most resultative one based on the N50, and a percentage of the assembly made up of scaffolds larger than 200,000/300,000 bp.

### 4.4. RAD-seq Data Analysis Pipelines and Construction of SNP Dataset

Variant calling using RAD-seq data is known to be complicated by several notable biases [16], and the resulting sets of variants may contain numerous genotyping errors. Hence, we decided to apply both general-purpose variant calling pipelines (based on the Genome Analysis ToolKit Haplotype Caller (GATK-HC) v.4.2.1.0 [15], as well as two software tools designed to work with GBS data—TASSEL5 v. 5.2.70 [19] and NGSEP v. 4.1.0 [18].

For GATK-HC and NGSEP pipelines, raw reads were demultiplexed using NGSEP. For GATK-HC analysis, reads were then aligned onto a reference genome assembly using BWA MEM v. 0.7.17 [37]. Aligned reads were processed with GATK in the GVCF mode, the resulting per-sample GVCF files were combined and jointly genotyped using GATK. For NGSEP, demultiplexed reads were processed using an error correction algorithm in NGSEP. Error corrected reads were aligned onto the reference genome using NGSEP’s built-in read alignment method. Resulting alignment files were used for multisample variant discovery. For TASSEL 5, the TASSEL GBSv2 pipeline was utilized. Raw reads were imported into the TASSEL GBS database to construct the tag sequences. Tags were then aligned onto the reference genome using Bowtie2 [38]. Aligned tags were loaded into the GBS database and used for discovery variant calling. Quality statistics for variant calls from all three tools were collected using a custom Python script (minor allele frequency, site-level heterozygosity, allele balance (fraction of reads with non-reference allele for heterozygous genotypes), and depth at variant site was evaluated.

Due to substantial differences in both the number of discovered variants (8097 for TASSEL, 103,413 for GATK, and 358,446 for NGSEP) and the main quality metrics (MAF, heterozygous genotype count, allele balance, and depth at variant site (Appendix A)), we decided to aggregate variant calls produced by different tools. As the proportion of shared calls between pairs of variant calling pipelines was also low (Appendix A), and only 2625 variants were successfully identified by all three variant callers (Appendix A), we selected variant sites that were discovered by GATK-HC and at least one of the GBS-specific tools (NGSEP and/or TASSEL) using a custom script in Python. When merging the call sets, shared sites were identified by variant position, irrespective of the alternative allele. GATK-based genotypes were used for further analysis for quality reasons (see Results). The combined variant dataset was pre-processed using Hail v. 0.2 (https://github.com/hail-is/hail, accessed on 24 August 2021). Variant quality statistics were computed using the Hail built-in functionality. Variants with variant call rate (the proportion of genotyped samples) < 0.75, observed heterozygosity > 0.5, mean per sample genotype quality (GQ) < 15, or quality-to-depth (QD) ratio < 20 were excluded. Filtered variants were used to compute the sample quality metrics, and samples with high heterozygous-to-homozygous genotype (HET/HOM) ratio > 2 or call rate (the proportion of genotyped variants) < 0.8 were removed. Finally, variants with minor allele frequency (MAF) > 0.05 in the filtered dataset were selected for further analyses.

### 4.5. Population Structure Analysis

Population stratification was analyzed using the principal component analysis (PCA) with Hail v.0.2. For ADMIXTURE [20] analysis, variants were LD-pruned using Hail (independent variants at r^2^ threshold < 0.2 were selected) and pre-processed using a custom script followed by conversion to the PLINK v.2.0 [39] binary format. ADMIXTURE was run using several values of K (number of presumed ancestral populations) ranging from 2 to 6. ADMIXTURE results were plotted using a custom R script.

The inbreeding coefficient for each variant site in the final dataset was computed as follows:(1)$$F=1-\frac{Ho}{He}$$
where *Ho* is the observed heterozygosity (the proportion of heterozygous genotypes per variant site), and *He* is the expected heterozygosity computed using the Hardy–Weinberg equilibrium.

For linkage disequilibrium analysis, pairwise r^2^ scores between SNPs located within the 10-Mb windows were computed using Hail v. 0.2.

### 4.6. Phenotypic Data Preprocessing

For each trait, the distribution of the values was examined, and points deviating more than 3 standard deviations from the median were removed. The raw values were then processed using the Inverse Rank-based Normalization (INT), a method that has been shown to increase the statistical power in genome-wide association studies [40]. To analyze the correlation between traits, Pearson’s correlation was computed using the normalized trait values.

### 4.7. Genome-Wide Association Analysis

Genome-wide association analysis was performed using Hail v. 0.2. or fixed and random model Circulating Probability Unification (FarmCPU) [21]. A generalized linear model (GLM) was used in Hail, with the first two principal components used as covariates. For FarmCPU analysis, genotypes from the VCF file were processed using a custom script for data formatting. For both methods, the genome-wide association results were evaluated using quantile–quantile plots, and the genomic inflation factor (λGC) was computed. Genome-wide association results were visualized using the CMplot packages (https://github.com/YinLiLin/CMplot) (Accessed on 30 September 2021).

For functional annotation of the genome-wide associations, the reference transcriptome assembly generated using Trinity V. 2.8.5 [41] was aligned onto the reference genome assembly using minimap2 v.2.2.22 [42].

### 4.8. SNP Validation

To validate SNPs in the final call set, the PCR-based assays were designed for those SNPs that were highlighted by the GWAS (Table 1). The PCR primers were designed in the following way: first, each PCR primer was constructed using the PrimerQuest software (https://eu.idtdna.com/pages/tools/primerquest?returnurl=%2FPrimerquest%2FHome%2FIndex, accessed on 24 August 2021) with default settings. Then, primers were aligned against the reference genome using Bwa V. 0.7.3. to prove that they do not anneal anywhere else outside the target contig. For such an alignment, the minimum sequence identity of 85% was used as the threshold value. Next, the designed primers were further checked via primer design settings of UGENE v. 39.0. using the corresponding contig as a target sequence. Only primers with the unique sequence within the target contig were used for the downstream analysis. Next, PCR validations for 6 SNPs were performed by PCR with designed primers and further Sanger sequencing. The PCR mixture consisted of 2.5 μL of 10× Taq buffer with MgCl2, 1 μL of dNTPs (10 mM), 0.5 μL of each primer (10 mM), 0.5 μL of Taq polymerase (5 U/μL), and 2 μL of DNA (10–30 ng/μL). The volume was adjusted to 25 μL by deionized water. For primers gu59, gu229, gu182, gu37, and gu188, the amplification program was as follows: 1 cycle (3 min at 95 °C), 35 cycles (30 s at 95 °C, 30 s at 57 °C, 1 min at 72 °C), and 1 cycle (5 min at 72 °C). For primer gu88 the amplification program consists of the 1 cycle (3 min at 95 °C), 35 cycles (30 s at 95 °C, 30 s at 59 °C, 1 min at 72 °C), and 1 cycle (5 min at 72 °C).

For sequencing, the PCR products were purified using AMPure XP magnetic beads according to the Agencourt AMPure XP protocol. Preparation of probes for sequencing was carried out with BigDye Terminator v3.1 (Thermo Fisher Scientific). The required concentration of the PCR product was adjusted according to the BigDyeTM Terminator v3.1 Cycle Sequencing Kit User Guide. The PCR program and further purification of the PCR product using 125 mM EDTA were carried out in accordance with the manufacturer’s recommendations.

The SNP information is available from Appendix A Appendix A, where the columns “upstream sequence” and “downstream sequence” provide information about reference genome sequence flanking SNPs.

## Figures and Tables

**Figure 1 plants-10-02063-f001:**
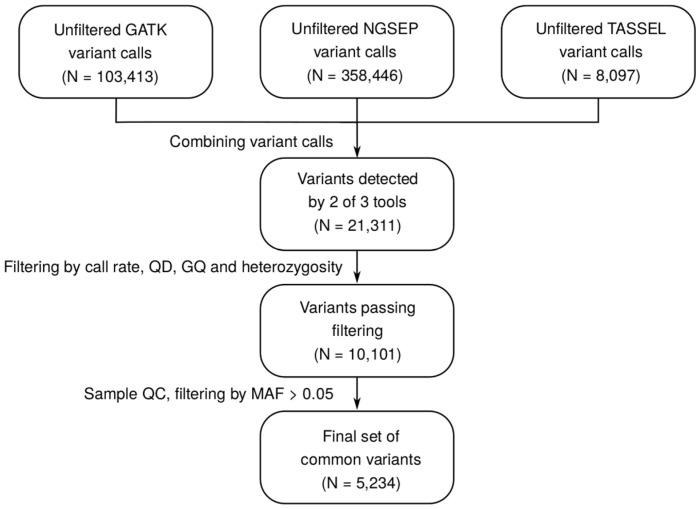
A workflow diagram describing the construction of a reference variant dataset for guar (*Cyamopsis tetragonoloba* (L.) Taub.).

**Figure 2 plants-10-02063-f002:**
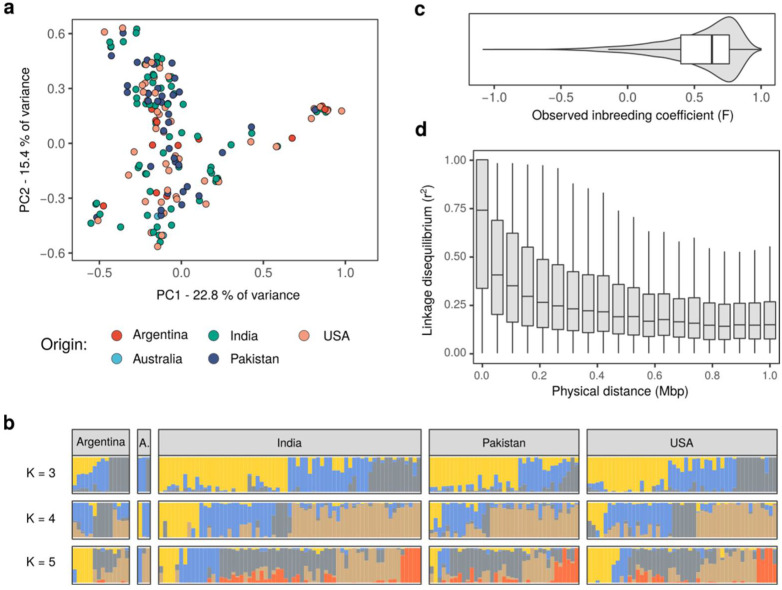
Reference genotype panel identifies the genetic structure of the guar populations. (**a**) Principal component analysis of the guar genotype dataset. Each point on the plot corresponds to one sample. Color of the points corresponds to the origin of the sample. (**b**) Bar plots representing results of the ADMIXTURE analysis for K = 3 (top), K = 4 (middle), and K = 5 (bottom). Color of the bar corresponds to one of K ancestral populations, height of the colored bar corresponds to the relative contribution of the ancestral population to the genotype of each sample. (**c**) Violin plot and box plot representing the distribution of the Inbreeding coefficient values computed for each variant in the guar genotype dataset. (**d**) Boxplots representing the linkage disequilibrium (r^2^) values for pairs of SNPs located at indicated physical distance in the same contig of the reference genome assembly.

**Figure 3 plants-10-02063-f003:**
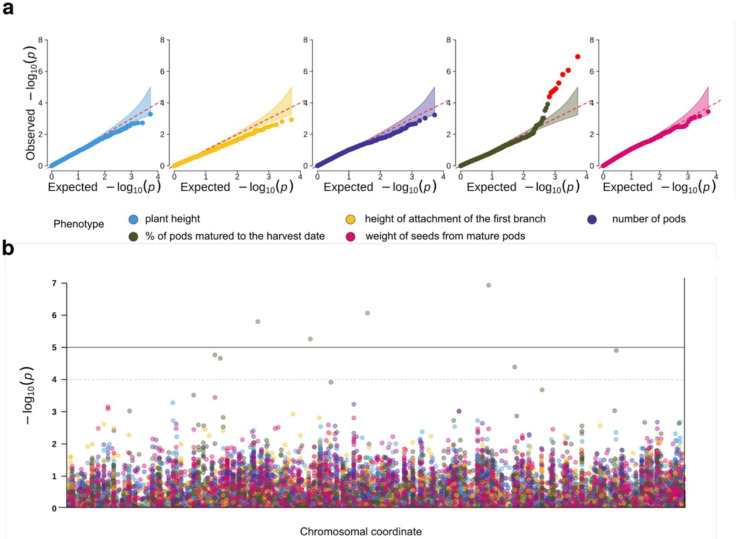
Genome-wide association analysis of productivity-related traits in guar. (**a**) Quantile–quantile plots for the association *p*-values against indicated traits. Envelope corresponds to the 95% confidence interval for the expected value. Variants reaching significance level of * p*< 1 × 10^−4^ are highlighted in red. (**b**) Manhattan plot showing association *p*-values for all indicated traits. Thresholds correspond to the call set-wide significance (bold line, 1 × 10^−5^) and sub-significance threshold (dashed line, 1 × 10^−4^). Colors of the points correspond to traits.

**Table 1 plants-10-02063-t001:** Variants associated with the percentage of mature beans in guar at *p*-value < 1 × 10^−4^.

Variant Position	*p*-Value *	MAF	Function of the Nearest Transcript Match in *A. thaliana*	Function of the Nearest Transcript Match in *G. max*
Contig_59:14389378	1.2 × 10^−7^	0.23	MYB domain protein, MYB43	Transcription factor MYB20
Contig_37:351180	8.5 × 10^−7^	0.36	ANU1, SECA2, Component of the thylakoid-localized Sec system	SECA2, Protein translocase subunit
Contig_229:330388	1.6 × 10^−6^	0.22	Single hybrid motif superfamily protein	Glycine cleavage system H protein 2
Contig_88:4931265	1.3 × 10^−5^	0.33	PGSIP7, plant glycogenin-like starch initiation protein 7	PIGSIP7/PIGSIP8 Putative glucuronosyltransferase
Contig_182:841660	1.7 × 10^−5^	0.21	ABCG40, ATP-Binding cassete G40 **	ABC transporter G family member 39 **
Contig_188:3026232	2.2 × 10^−5^	0.06	RAE1, Regulation of ATALMT1 expression	F-box/LRR-repeat protein 3

*****—*p*-values correspond to the Fixed and Random effect model (FarmCPU); **—exact transcript match.

## Data Availability

All of the mentioned data available in supplementary materials. Code available at https://github.com/mrbarbitoff/guar_snv, accessed on 24 August 2021.

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
