# Peer review of "Development of SNP Set for the Marker-Assisted Selection of Guar (Cyamopsis tetragonoloba (L.) Taub.) Based on a Custom Reference Genome Assembly"

_plants, 2021, doi:10.3390/plants10102063_

Round 1

Reviewer 1 Report

Authors have developed the SNV set for the marker-assisted selection of guar (Cyamopsis tetragonoloba (L.) Taub.) based on a custom reference genome assembly. They report a set of 4907 common SNPs and 327 InDels, generated from RADseq genotyping data of 166 guar plants of different geographical origin. Apart from this authors have also assembled the reference genome of this plant to detect SNVs. Based on the variant identification, they reported several important genes and compiled the molecular markers of important agro biological traits. I request authors to address the following comments.

  1. Chromosome scale reference genome is already available (https://www.biorxiv.org/content/10.1101/2020.05.16.098434v1) for (Cyamopsis tetragonoloba (L.) Taub.). why authors have not used this genome for the variant calling?
  2. Why authors have created custom reference genome, when good reference already available?
  3. Which reads were used for the variant calling, Illumina reads or Oxford NT reads?
  4. Language editing required.

Author Response

Dear Reviewer, 

Thank you very much for your valuable comments. Below, we are addressing them in detail:

  • Chromosome scale reference genome is already available (https://www.biorxiv.org/content/10.1101/2020.05.16.098434v1) for (Cyamopsis tetragonoloba (L.) Taub.). Why authors have not used this genome for the variant calling?

       2 Why authors have created custom reference genome, when good reference already available?

The preprint you mentioned has not yet been published and, therefore, data of the genome assembly is still not available from NCBI. To make the point clear we made an addition to the Introduction section:

Since a high-quality reference genome assembly of guar [9] is still not publicly available, we constructed a custom assembly using both short- and long-read sequencing technologies.”

Unfortunately, we are not ready to make accessible the custom reference genome assembly, since we are currently working on its proper annotation. As soon as we finish this work, the assembly of guar genome will be deposited in NCBI.

  1. Which reads were used for the variant calling, Illumina reads or Oxford NT reads?

In the new version of the manuscript we clarified this point by adding to the corresponding Results (section 2.1) the following modified sentence:  

“For the SNP calling procedure two RAD-Seq libraries each containing 96 samples with unique barcodes were sequenced using the Illumina HiSeq2500 platform…”

  1. Language editing required

The language has now been thoroughly checked.

Reviewer 2 Report

This study deals with a crop (guar) that has great potential and importance for the countries of origin but also for new countries. For this reason, this study is very interesting and important. In this paper, 4907 common SNPs were reported along with 327 InDels using RADseq genotyping data of guar plants of different geographical origin and nature. This study is heavy in experiments and work done. A custom guar reference genome was assembled and used to enhance the robustness of the detected SNVs. Three bioinformatic pipelines were used to ensure the most accurate set of SNVs. A genome-wide association study was performed and six markers were linked with important agronomic traits. Hence, we believe that the developed SNV set allows for identification of confident molecular markers of important agrobiological traits. This study is therefore an important starting point for MAS in breeding of this crop, but still more testing in the field and more GWAS studies are needed.

The scientific part of the study is very solid and the authors have described it in detail. My suggestions for improvement concern only the presentation quality of the manuscript. English must be improved and a few syntax errors must be fixed. In the appearance of the manuscript, several gaps exist between words, sentences and paragraphs that are ugly. Some tables are cut from page breaks. Some more details that need fixing:

  • A reader may not know what is a SNV, maybe an explanation is needed in the Introduction.
  • The last sentences of the introduction may well be the conclusions of the study.
  • What is the meaning of an inbreeding coefficient and the expected heterozygosity or any comparison with HW conditions in a set of genotypes that are not a population (Figure S1 also)? Also, any set of genotypes mixed from different populations would result in a homozygote excess, due to the Wahlund effect, so the results of this study were expected. For this reason, this analysis does not add any information.
  • Some parts of "Results" repeat the same parts from "Materials and Methods" and the beginning of the Discussion do the same with the Results.

In overall, this study is suited for this journal, after fixing these minor issues.

Author Response

Dear Reviewer, 

Thank you very much for your valuable comments. Below, we are addressing them in detail:

 The scientific part of the study is very solid and the authors have described it in detail. My suggestions for improvement concern only the presentation quality of the manuscript. English must be improved and a few syntax errors must be fixed. In the appearance of the manuscript, several gaps exist between words, sentences and paragraphs that are ugly. Some tables are cut from page breaks. 

English has been thoroughly checked. Formatting, syntax errors and gaps between words are fixed as best as we could.

Some more details that need fixing:

  • A reader may not know what is a SNV, maybe an explanation is needed in the Introduction.

Thanks for the comment! We realized that the use of the abbreviation SNV in our manuscript was incorrect. According to the Garvan Institute of Medical Research, an SNV is actually considered as a SNP, if the variant is present in at least 1% of the population. Since we used MAF> 5% as a filtering criterion, all of our claimed nucleotide substitutions are actually SNPs. Thus, we have replaced SNV with the term SNP in the manuscript.

  • The last sentences of the introduction may well be the conclusions of the study.

The  last sentence of the introduction was removed.

  • What is the meaning of an inbreeding coefficient and the expected heterozygosity or any comparison with HW conditions in a set of genotypes that are not a population (Figure S1 also)? Also, any set of genotypes mixed from different populations would result in a homozygote excess, due to the Wahlund effect, so the results of this study were expected. For this reason, this analysis does not add any information.

We absolutely agree that homozygote excess is expected for guar, due to obligate autogamy of the species, and, possibly, due to the Wahlund effect. The point is that the bioinformatic-based SNPs published so far for guar (e.g. Indian J. Genet.,80(2) 179-185 (2020), DOI: 10.31742/IJGPB.80.2.8) show ~50% of heterozygosity for the discovered SNPs, which is hardly likely for self-pollinating species. In our experience, the high level of heterozygosity is usually due to errors when aligning Illumina reads to the reference genome. We discuss the pitfalls in the "Discussion" section. For this reason, we would prefer to keep Figure S1 (in the new ms version this is Figure S2)  to demonstrate homozygote excess as a proof of robustness of the developed SNPs. 

We also include an additional sentence in 2.2. section to clarify that the observed genotype frequency distribution can be partially explained by specific properties of our sample:

“...While increased homozygosity of samples might also reflect the nature of the sample (a mixture of samples of different origin rather than a random sample from a single population), the genotype distribution confirms known aspects of guar biology and further confirms the general accuracy of genotyping…”

  • Some parts of "Results" repeat the same parts from "Materials and Methods" and the beginning of the Discussion do the same with the Results.

The ‘Result’ and ‘Materials and Methods’ sections have been completely reorganized to avoid repeating information. We also edited Figure 1, by moving some parts of the Figure into Supplementary Materials.

Reviewer 3 Report

I appreciate the authors' efforts in developing SNP markers for a percentage of pods matured to the harvest data, a key agronomic trait although concrete proof linking this traits with particular SNP would have been more interesting. Despite the odds, this topic is interesting to the readers. I have few concerns, May I request the authors to address this, if possible?.

  1. Is the genomic information/sequence data made from this has been submitted to any of the public databases? If not, it can be deposited
  2. Line 99 should reveal which genome and what is the version and when it is accessed
  3. Also for all bioinformatics tools, access date should be mentioned
  4. Line 30, what do you mean earliness of guar plants?......meant flowering?
  5. Importantly, section 2.5 should be revisited, with more information about results than a description of methods should be provided. Resulted should be elaborated rather than saying successfully validated
  6. line 475, different font/ size, please check carefully
  7. Line 321, the description should be added to the other five SNPs
  8. Line 324-326, How do you relate the SNP and gene expression pattern?
  9. Line 315, What do you mean by a strong genome-wide association for these traits? Can you provide the percentage of relatedness? I hope my clarifications and suggestions will improve your manuscript depth
  10.  

Author Response

Dear Reviewer, 

Thank you very much for your valuable comments. Below, we are addressing them in detail:

  • Is the genomic information/sequence data made from this has been submitted to any of the public databases? If not, it can be deposited

We believe that the main results of our study are present in the Supplementary Materials Table S2, containing information about 4907 SNPs and 327 InDels, we propose to use as molecular markers in other guar studies. To support their further use, the Table S2 provides information about 150 bp upstream and 150 bp downstream sequences, flanking each polymorphic locus.

  • Line 99 should reveal which genome and what is the version and when it is accessed

In the new version of the manuscript we clarified that we used a custom version of guar reference genome by adding to the ‘Introduction’ section the following explanation: “Since a high-quality reference genome assembly of guar [9] is still not publicly available, we constructed a custom assembly using both short- and long-read sequencing technologies.” 

Unfortunately, we are not ready to make accessible the custom reference genome we used for SNP calling, since  we are currently working on its proper annotation. We hope to publish another paper about guar genome soon, and then the assembly of the genome will be deposited in NCBI.

  • Also for all bioinformatics tools, access date should be mentioned.

For each bioinformatics tool, we added the version number we used. However, to avoid repeating the same information, software version numbers have been indicated only in the ‘Material and Methods’ section.

  • Line 30, what do you mean earliness of guar plants?......meant flowering?

On line 30 we have replaced ‘earliness’ with ‘early flowering”. The sentence now reads: “Earlier we suggested that genes involved in myo-inositol phosphate metabolism have significant impact on early flowering of guar plants.”

  • Importantly, section 2.5 should be revisited, with more information about results than a description of methods should be provided. Resulted should be elaborated rather than saying successfully validated

Section 2.5 (in the new version of manuscript 2.4) is rewritten and renamed. Currently,  this section claims “PCR validation of the bioinformatics-based SNP prediction”.  We explain that the main task of the validation experiment was to confirm by PCR and Sanger sequencing the physical presence of target nucleotide substitutions in those guar plants, where corresponding SNPs were predicted using a bioinformatics-based approach. We explain how we selected a subset of genotypes from 166 plants with predicted alternative alleles, sequenced the PCR products, and confirmed the presence of SNPs exactly where they were expected. We also discuss  the correlation of between SNP and a trait, and recommend best options for marker assisted breeding.

  • line 475, different font/ size, please check carefully

On Line 475 the sentence “Genome-wide association analysis was performed using Hail v. 0.2. or fixed and random model Circulating Probability Unification (FarmCPU)” is checked.

  • Line 321, the description should be added to the other five SNPs

Line 321 stated that “... Among six SNVs associated with the percentage of pods matured to the harvest date just one (contig_182:841660, Table 1) was found inside the putative transcript sequence.” Thus, out of 6 SNPs, only one SNP can be described in detail, since we know the gene containing this SNP.

For other SNPs, we tried to identify the nearest transcript match. This was not always possible. So, Line 347 continues: “Two other associated SNPs (contig_59:14389378 and contig_188:3026232) were discovered near the guar coding sequences, showing homology to transcription factor 348 MYB20 and F-box/LRR-repeat protein 3 respectively (Table 1). 

  • Line 324-326, How do you relate the SNP and gene expression pattern?

To make the point clear, we rephrase the explanation as follows (insertion is underlined):  “ Therefore, a direct relationship can be assumed between the increased expression level of the AT1G15520 orthologous gene of guar and early maturing of guar plants. Since the structure of the corresponding gene in guar has not yet been investigated, there is no option to predict any amino acid substitutions caused by the discovered SNP. Nevertheless, we can assume the presence of tight linkage disequilibrium between SNPs in the gene exon and other polymorphisms in its promoter region that affect gene expression.”

  • Line 315, What do you mean by a strong genome-wide association for these traits? Can you provide the percentage of relatedness?

To clarify what means the strong genome-wide association established for the key agrobiological trait “(percentage of pods matured to the harvest day), we have modified the corresponding sentence as follows (underlined): “ Thus, the earliness of the pods maturation assessed as percentage of pods matured to the harvest date is of the greatest importance among the yield-contributing traits. Fortunately, a strong genome-wide association signal (6 variants with p < 0.0001, Table 1) was discovered for this key agrobiological trait.”

Reviewer 4 Report

The manuscript 'Development of SNV set for the marker assisted selection of guar based on a custom reference genome assembly' prepared by Grigoreva et al. described a SNV maker associated with different trait use RAD seq genotyping data. This method will generate more and higher quality data for genotyping and assembly. Different bioinformatic protocol were applied in the analysis and their results were compared as well. Finally, the authors identified six interested SNP markers that can determine associated with different genes were validated. However, there are several major/minor concerns should be addressed before publication.

1) In the abstract and the first half of introduction. Please indicate what the  'SNV' stands for. "Single nucleotide variant' is not widely used term as 'SNP'.

2) In result section, the major issue is a lot of description is repeated with the Materials and Methods section. For example, L103-109, L115-135. L227-237, etc. I strongly suggest authors can re-organize and concise this section. I can understand you prefer to compare the different parameters/filters. However, too much repeats will make the manuscript redundant.

3) You did structure analysis of the involved germplasms. Have you done any phylogenetic analysis as well to check the population evolution and divergence. 

4) GWAS were conducted to identify the candidate SNPs/clusters that highly associated with plant height, pod traits, seed weight etc.. If I am correct, you just do the genome/transcriptome scan of the LD/SNPs region with percentage of mature pods and find six candidate SNPs. How about others? I do not think, adding the analysis of others is a huge work in the same manner. I will suggest authors could add the SNPs with other included traits as well after GWAS.

5) PCR validation, the results presents here is not sufficient. For example, did you find the correlation of between each SNP and traits. Have you done the validation in the population used in GWAS. Finally, a conclusion such as  SNP 'A'  or 'C' (example) indicate the certain traits. Among the six, which one is the best option for marker assisted breeding program.

Another suggestion you may consider, expression analysis of these gene in certain distinct plant genotypes will also helpful. This will also benefit to your discussion in 'Discussion' section.

Author Response

Dear Reviewer, 

Thank you very much for your valuable comments. Here, we are addressing them in detail:

1) In the abstract and the first half of introduction. Please indicate what the  'SNV' stands for. "Single nucleotide variant' is not widely used term as 'SNP'.

Thanks for the comment! We realized that the use of the abbreviation SNV in our manuscript was incorrect. According to the Garvan Institute of Medical Research, an SNV is considered as a SNP if the variant is present in at least 1% of the population. Since we used MAF> 5% as a filtering criterion, all the nucleotide substitutions we declared are in fact SNPs. Thus, we have replaced SNV with the term SNP in the manuscript.

2) In result section, the major issue is a lot of description is repeated with the Materials and Methods section. For example, L103-109, L115-135. L227-237, etc. I strongly suggest authors can re-organize and concise this section. I can understand you prefer to compare the different parameters/filters. However, too much repeats will make the manuscript redundant.

Done. The ‘Result’ and ‘Materials and Methods’ sections have been completely re-organized to avoid repeating information. We also edited Figure 1, by moving some parts of the Figure into Supplementary Materials.

3) You did structure analysis of the involved germplasms. Have you done any phylogenetic analysis as well to check the population evolution and divergence. 

To reveal divergence within the germplasm analysed we performed Principal Component Analysis and the genetic clustering algorithm, ADMIXTURE (lines 172-173). So, we analysed divergence of genotypes, but not in the context of evolution. Analyzing intraspecific diversity, we did not carry out phylogenetic analysis, focusing mainly on discovery of robust SNPs and their potential value for MAS of guar.

4) GWAS were conducted to identify the candidate SNPs/clusters that highly associated with plant height, pod traits, seed weight etc.. If I am correct, you just do the genome/transcriptome scan of the LD/SNPs region with percentage of mature pods and find six candidate SNPs. How about others? I do not think, adding the analysis of others is a huge work in the same manner. I will suggest authors could add the SNPs with other included traits as well after GWAS.

We performed the association analysis for all the traits that we have phenotyped, however the highly associated SNPs were detected just for one trait - percentage of mature pods. To make the point clear we rephrase the text in section 2.3 ( the insertions are underlined):

“ We applied two types of association analysis models to detect genome-wide association across our panel of phenotypes (e.g. plant height, height of the first branch attachment, total number of pods, percentage of pods matured to the harvest date and total weight of seeds harvested from mature pods). GLM-based association testing did not identify any significant genome-wide associations (Supplementary Figure S5); however, significant genomic inflation was observed in the association statistics only for one of the traits (percentage of pods matured to the harvest day).”

5) PCR validation, the results presents here is not sufficient. For example, did you find the correlation of between each SNP and traits. Have you done the validation in the population used in GWAS. Finally, a conclusion such as  SNP 'A'  or 'C' (example) indicate the certain traits. Among the six, which one is the best option for marker assisted breeding program.

Section 2.5 (currently 2.4) is rewritten and renamed. In the new version of the manuscript,  this section 2.4 claims “PCR validation of the bioinformatics-based SNP prediction”.  We explain that the main task of the validation experiment was to confirm by PCR and Sanger sequencing the physical presence of target nucleotide substitutions in those guar plants, where corresponding SNPs were predicted using a bioinformatics-based approach. We explain how we selected a subset of genotypes from 166 plants with predicted alternative alleles, sequenced the PCR products, and confirmed the presence of SNPs exactly where they were expected. We also discuss  the correlation of between SNP and a trait, and recommend best options for marker assisted breeding.

Another suggestion you may consider, expression analysis of these gene in certain distinct plant genotypes will also helpful. This will also benefit to your discussion in 'Discussion' section.

We would prefer to study the candidate gene more deeply in the frame of a separate study.

Round 2

Reviewer 3 Report

Dear Authors

Thanks for the response. Congratulations on the publication.

Reviewer 4 Report

Dear authors,

Thanks for your efforts in revising the manuscript. I am glad to see that the revised version improved significantly and addressed my concerns as well. I will expected to see the manuscript can be published soon. A gently remind, since you did a lot of changes/editing, a careful proof reading is highly required before the manuscript publishing online.